# The Impact of Cytomegalovirus Infection on Natural Killer and CD8+ T Cell Phenotype in Multiple Sclerosis

**DOI:** 10.3390/biology13030154

**Published:** 2024-02-28

**Authors:** Valentina Perri, Maria Antonella Zingaropoli, Patrizia Pasculli, Federica Ciccone, Matteo Tartaglia, Viola Baione, Leonardo Malimpensa, Gina Ferrazzano, Claudio Maria Mastroianni, Antonella Conte, Maria Rosa Ciardi

**Affiliations:** 1Department of Public Health and Infectious Diseases, Sapienza University of Rome, 00185 Rome, Italy; mariaantonella.zingaropoli@uniroma1.it (M.A.Z.); patrizia.pasculli@uniroma1.it (P.P.); federica.ciccone@uniroma1.it (F.C.); claudio.mastroianni@uniroma1.it (C.M.M.); maria.ciardi@uniroma1.it (M.R.C.); 2Department of Human Neurosciences, Sapienza University of Rome, 00185 Rome, Italy; matteo.tartaglia@uniroma1.it (M.T.); viola.baione@uniroma1.it (V.B.); gina.ferrazzano@uniroma1.it (G.F.); antonella.conte@uniroma1.it (A.C.); 3IRCCS Neuromed, 86077 Pozzilli, Italy; leonardo.malimpensa@uniroma1.it

**Keywords:** cytomegalovirus, multiple sclerosis, NK cells, CD8+ T cells, NKT-like cells, NKG2C, flow cytometry

## Abstract

**Simple Summary:**

Natural killer cells and cytotoxic CD8+ T cells play a complementary role in controlling cytomegalovirus, a ubiquitous herpesvirus that establishes a lifelong latent infection in the host. Due to its ability to modulate the immune response, cytomegalovirus can impact the course of multiple sclerosis, an immune-mediated disease of the central nervous system. However, whether this herpesvirus may play a beneficial or a detrimental role in multiple sclerosis is currently uncertain. The cytomegalovirus seropositivity status increases both the expansion of highly differentiated CD8+ T cells and the number of natural killer cells expressing the NKG2C receptor. Given that NKG2C is an activating receptor that enhances natural killer cell-mediated cytotoxicity and recruitment of inflammatory cells, it is reasonable to investigate the involvement of this receptor in immune-mediated diseases. In the present study, we explored the magnitude of cytomegalovirus-induced immune changes in multiple sclerosis in order to increase the knowledge of the relationship between viruses and immune-mediated diseases.

**Abstract:**

Multiple sclerosis (MS) is a debilitating neurological disease that has been classified as an immune-mediated attack on myelin, the protective sheath of nerves. Some aspects of its pathogenesis are still unclear; nevertheless, it is generally established that viral infections influence the course of the disease. Cytomegalovirus (CMV) is a major pathogen involved in alterations of the immune system, including the expansion of highly differentiated cytotoxic CD8+ T cells and the accumulation of adaptive natural killer (NK) cells expressing high levels of the NKG2C receptor. In this study, we evaluated the impact of latent CMV infection on MS patients through the characterization of peripheral NK cells, CD8+ T cells, and NKT-like cells using flow cytometry. We evaluated the associations between immune cell profiles and clinical features such as MS duration and MS progression, evaluated using the Expanded Disability Status Scale (EDSS). We showed that NK cells, CD8+ T cells, and NKT-like cells had an altered phenotype in CMV-infected MS patients and displayed high levels of the NKG2C receptor. Moreover, in MS patients, increased NKG2C expression levels were found to be associated with higher EDSS scores. Overall, these results support the hypothesis that CMV infection imprints the immune system by modifying the phenotype and receptor repertoire of NK and CD8+ T cells, suggesting a detrimental role of CMV on MS progression.

## 1. Introduction

Multiple sclerosis (MS) is a multifactorial demyelinating and neurodegenerative disorder of the central nervous system (CNS); the deregulation of the immune response plays a crucial role in its pathophysiology [1,2]. Genetic and environmental factors such as viral infections are strongly linked to MS susceptibility and progression [3]. Viruses can cause damage directly by infecting the CNS or indirectly through modifications of the immune response, and several authors have focused on herpesvirus involvement for its ability to induce a persistent latent infection in the hosts [4,5]. The Epstein–Barr virus (EBV) appears to play a major role in MS; however, many studies have also focused on the putative involvement of cytomegalovirus (CMV) [6,7]. There is controversy regarding the role of CMV in MS pathogenesis. Some studies reported that CMV is associated with a lower MS susceptibility [8], while others suggested that CMV may contribute to the exacerbation of MS [9,10]. CMV is a ubiquitous beta herpesvirus that infects 40–80% of adults in industrialized countries and the entire population in developing countries [11,12]. Usually, the primary infection is asymptomatic, but some individuals develop a mononucleosis-like disease. Once infected, the virus persists in the host for life as a latent infection in undifferentiated cells of the myeloid lineage, such as CD34+ hematopoietic progenitor cells [13]. Natural killer cells (NK) and CD8+ T cells are the main effector cells that play a crucial role in the defense against CMV [14,15]. 

NK cells are innate lymphoid cells that recognize and eliminate CMV-infected cells. CMV has a complex relationship with NK cells: it plays a crucial role in shaping the NK cell receptor repertoire and drives the expansion of the population of NK cells with memory-like features. In humans, memory-like NK cells are commonly associated with the expression of NKG2C [16]. NKG2C is an activating NK cell receptor of the C-type lectin superfamily that binds the human non-classical MHC class I molecule HLA-E on the cell surface [17,18]. NKG2C-positive NK cells display higher cytotoxic capacity, elevated sensitivity to cytokines, such as interleukin (IL)-12 and IL-18, and increased ability for interferon (IFN)-γ production compared to their NKG2C-negative counterparts [19,20]. The mechanisms by which CMV drives the expression of NKG2C on NK subsets are unknown; however, it is well established that NKG2C+ cells play a role in controlling CMV infection, both in vitro and in vivo [21]. Both human studies and animal models of MS support the role of NK cells in the pathogenesis of this disease [22,23]. However, whether the interaction between NKG2C and HLA-E plays a role in the pathogenesis of MS has been evaluated only in one in vitro study [24]. 

CD8+ T cells play a critical role in the immune response to viral infections. Persistent CMV infection represents a major contributor to the senescence of CD8+ T cells [25]. Specifically, CMV promotes the accumulation of late-differentiated CD8+ T cells characterized by replicative senescence, the inability to proliferate in response to different triggers, the loss of the CD28 co-stimulatory marker, and the acquisition of the CD57 receptor [26,27]. Quantitative changes in the late-differentiated CD8+ T cell population are observed in MS and other immune-mediated diseases [28]. 

Considering the profound impact that CMV exerts on the immune system, a putative influence in MS immunopathology is conceivable. However, it is currently unclear whether CMV through these immune changes may have a beneficial or a detrimental effect on MS. In this research, we explored the impact of CMV on MS through a characterization of peripheral NK cells and T cells to provide a better understanding of CMV-associated immune cell alterations and to investigate CMV putative involvement in MS.

## 2. Materials and Methods

### 2.1. Study Population

At the Neuroinfectious Unit of Policlinico Umberto I Hospital, Sapienza University of Rome, we enrolled MS patients. For each patient, blood samples were collected before starting disease-modifying therapies (DMTs). The following parameters were evaluated: sex, age, disease duration, and disease progression, the latter through the EDSS (Expanded Disability Status Scale), which scores disease progression from 0 to 10 in 0.5-unit increments, with higher values representing higher levels of disability [29]. We also enrolled an age- and gender-matched control population that included healthy donors (HDs). From each subject, a total of two samples were collected: one sample for plasma and cells, and the other for serum. The study was approved by the Ethics Committee of Policlinico Umberto I, Sapienza University of Rome (protocol numbers 130/13 and 353/20), and all subjects signed an informed consent form before blood collection.

### 2.2. CMV Serology

A standard serological diagnostic analysis was performed to evaluate specific CMV-IgG and CMV-IgM antibodies. The serostatus was determined by the Liaison^®^ CMV IgG, IgM assay (DiaSorin, Saluggia, Italy), and the results are expressed as index value. 

### 2.3. Detection of CMV DNA by Real-Time PCR

Real-time PCR was performed to reveal CMV DNA in the plasma samples. Viral DNA was extracted from 200 μL of plasma by the use of High Pure Viral Nucleic Acid Kit (Roche Biochemicals, Mannheim, Germany). The purified DNA was eluted and stored at −20 °C until use. For real-time PCR, the following primers and probes (TIB MOLBIOL, Berlin, Germany) were mixed with 2 μL of template DNA, 4 mM MgCl2, and 4 μL of LightCycler-FastStart DNA (Master Hybridization Probes kit; Roche Biochemicals, Mannheim, Germany): 0.4 μM forward primer, 0.4 μM reverse primer, 0.2 μM fluorescein hybridization probe, and 0.2 μM LC-Red 640 probe. The sequences of the used specific primers for the *glycoprotein B gene* (254 bp) of CMV were previously reported (GenBank accession no. A13758). The hybridization probe sequences (5′-3′) comprised the donor fluorophore probe (CGTTTCGTCGTAGCTACGCRTACAT-fluorescein) and the acceptor fluorophore probe (LC-Red 640-ACACCACTTATCTYCTGGGCAGC-phosphate). The primers and probes were produced by TIB MOLBIOL (Berlin, Germany). The real-time PCR assays were run in the LightCycler^®^ 2.0 instrument (Roche Applied Science, Penzberg, Germany) using the following protocol: 10 min at 95 °C, followed by 45 cycles of 10 s denaturation at 95 °C, 10 s annealing at 58 °C, and 12 s extension at 72 °C. The melting temperature for the probe set was 59.2 °C.

### 2.4. Isolation of Peripheral Blood Mononuclear Cells

As previously described [30], peripheral blood mononuclear cells (PBMCs) were isolated from whole blood samples collected in ethylenediaminetetraacetic acid (EDTA) tubes by density gradient centrifugation (Histopaque^®^ 1077-1, Sigma-Aldrich, Saint Louis, MO, USA). The isolated PBMCs (1 × 106) were resuspended in 90% heat-inactivated fetal bovine serum (FBS) (Sigma-Aldrich, Saint Louis, MO, USA) and 10% dimethyl sulfoxide (DMSO) (Sigma-Aldrich, Saint Louis, MO, USA) and stored in liquid nitrogen.

### 2.5. Flow Cytometry Antibody Staining

We performed multiparameter flow cytometry to characterize the phenotype of NK cells, T cells, and NKT-like cells. As previously described [31], after thawing, PBMCs were incubated with specific mAbs in the dark for 30 min at 4 °C. To exclude dead cells from the analysis, a fixable viability dye (Zombie Aqua, BioLegend, San Diego, CA, USA) was included in the staining mixture. The following mAbs were used in the study: anti-CD3, pacific blue (PB)-conjugated (clone HIT3a), anti-CD16, phycoerythrin-cyanin7 (PE-cy7)-conjugated (clone 3G8), anti-CD8, allophycocyanin (APC)-conjugated (clone SK1), anti-CD57, PE-conjugated (clone HNK-1), anti-CD56, APC-cy7-conjugated (clone 5.1H11), anti-CD28, fluorescein isothiocyanate (FITC)-conjugated (clone CD28.2), anti-NKG2C, (CD159c), and peridinin–chlorophyll protein complex–cyanine5.5 (PerCP-Cy 5.5)-conjugated (clone 134591). All the mAbs were purchased from BioLegend (San Diego, CA, USA), except for anti-NKG2C, which was from BD Biosciences (Franklin Lakes, NJ, USA). 

### 2.6. Flow Cytometry Data Analysis

Flow cytometry was performed using the MACSQuant^®^ Analyzer 10 (Miltenyi Biotec, Bergisch Gladbach, Germany). By combining mAbs, we identified the following cell populations: NK cells (CD3− CD56+), CD56^bright^ NK cells (CD3− CD56++ CD16+/−), CD56^dim^ NK cells (CD3− CD56+ CD16+), CD57+ NK cells (CD3− CD56+ CD16+ CD57+), T cells (CD3+ CD56−), NKT-like cells (CD3+ CD56+), CD8+ T cells (CD3+ CD56− CD8+), early-differentiated CD8+ T cells (CD3+ CD56− CD8+ CD28+ CD57−), and late-differentiated CD8+ T cells (CD3+ CD56− CD8+ CD28− CD57+). The expression of the NKG2C receptor was evaluated on CD57+ NK cells by median fluorescence intensity (MFI). The analysis was performed using FlowJo (version 10.8.1) software. The gating strategy is shown in Appendix A.

### 2.7. Statistical Analysis

Statistical analysis was performed with GraphPad Prism 6.0 (GraphPad Software, San Diego, CA, USA). The Mann–Whitney U test was used to compare the medians of the two groups. Spearman correlation was used to evaluate the relationships between two continuous variables. The data from the experiments are expressed as median (range interquartile, IQR). Statistical significance was set at *p* < 0.05.

## 3. Results

### 3.1. Study Population

A total of 74 MS patients (39 males and 35 females) with a median age (IQR) of 51 (43–58) years were included in the study. At the time of enrollment, the median EDSS score (IQR) was 5.0 (3.0–6.0), while the median MS duration (IQR) was 11 (5–19) years. In addition, 18 HDs (7 males and 11 females) with a median age (IQR) of 52 (38–59) years were enrolled. Table 1 summarizes the demographic, clinical, and serological data of the 74 MS patients and 18 HDs included in the analyses.

### 3.2. Detection of Anti-CMV IgG Antibodies and CMV DNA in MS Patients

As reported in Table 1, the prevalence of anti-CMV IgG antibodies among the MS patients was 69%, specifically, 51% in males and 49% in females. No anti-CMV IgM antibodies, suggestive of recent primary infection, were detected. We found that the CMV serostatus in the MS patients was significantly associated with older age (*p* = 0.02). In contrast, the CMV serostatus was not associated with clinical variables such as MS duration and EDSS (Table 2). Real-time PCR was used to detect CMV DNA in plasma samples using specific primers and probe for a portion of the CMV glycoprotein B gene. Our results showed that CMV DNA was detected in the plasma samples of 16% of the MS patients. The patients who had detectable CMV DNA were CMV-seropositive. CMV DNAemia was not associated with aging or clinical variables (Table 2).

### 3.3. Increased NKG2C Expression Levels in MS Patients Compared to Healthy Donors

Overall, by flow cytometry, we analyzed 29 MS patients and 18 HDs. We compared the phenotype of NK cells T cells, NKT-like cells, and their subsets between MS patients and HDs (Table 3). Similar percentages of NK cells and NKT-like cells were observed in MS patients and HDs (Figure 1A and Figure 1F, respectively). Moreover, no differences were identified in the percentages of CD56^bright^, CD56^dim^, and CD57+ NK cells (Figure 1B, Figure 1C and Figure 1D, respectively). Conversely, the MS patients presented increased expression levels of NKG2C (*p* = 0.041) on the CD57+ NK cell subset compared to the HDs (Figure 1E). Moreover, no differences were observed in the percentages of CD8+ T cells, early-differentiated CD8+ T cells, and late-differentiated CD8+ T cells (Figure 1H, Figure 1I and Figure 1J, respectively). Conversely, the MS patients showed an increased percentage of total T cells (*p* = 0.001) compared to the HDs (Figure 1G).

### 3.4. The CMV Serostatus Influences the Peripheral Blood Phenotype in MS Patients

We subsequently evaluated the immunophenotype of peripheral NK, T, and NKT-like cells and their relative subsets in MS patients and HDs, according to their CMV serostatus (Table 4). We observed an increased percentage of CD57+ NK cells (*p* = 0.020) and NKT-like cells (*p* = 0.008) in CMV-seropositive MS patients compared to CMV-seronegative MS patients (Figure 2D and Figure 2F, respectively). Moreover, the expression level (MFI) of NKG2C on CD57+ NK cells (*p* = 0.008) was increased in CMV-seropositive MS patients compared to CMV-seronegative MS patients (Figure 2E). Conversely, the percentages of total NK cells and of the CD56^bright^ and CD56^dim^ NK cell subsets were comparable between CMV-seropositive MS patients and CMV-seronegative MS patients (Figure 2B and Figure 2C, respectively). Additionally, the expression level (MFI) of NKG2C (*p* = 0.022) was increased in CMV-seropositive MS patients compared to CMV-seropositive HDs (Figure 2E).

T cell immunophenotyping showed that the percentages of total T cells were comparable between CMV-seropositive MS patients and CMV-seronegative MS patients (Figure 2G). Also, for the CD8+ T cell subset, no statistically significant difference was found (Figure 2H). In contrast, we found a decreased percentage of the early-differentiated CD8+ subset (*p* = 0.023) and an increased percentage of the late-differentiated CD8+ subset (*p* = 0.002) in CMV-seropositive MS patients compared to CMV-seronegative MS patients (Figure 2I and Figure 2J, respectively). 

### 3.5. Correlations

Possible associations between the percentage of cell subsets and clinical parameters were investigated in MS patients. Figures show all the statistically significant correlations detected by the Spearman’s rank correlation coefficient (*p* < 0.05). A positive correlation between the MFI of NKG2C on CD57+ NK cells with both EDSS (ρ = 0.439, *p* = 0.022) and disease duration was observed (ρ = 0.423, *p* = 0.028) (Figure 3A and Figure 3B, respectively).

Moreover, possible correlations between cell subsets were studied. The late-differentiated CD8+ T cells percentage was positively correlated both with the CD57+ NK cell percentage (ρ = 0.435, *p* = 0.019; Figure 4A) and the MFI of NKG2C on CD57+ NK cells (ρ = 0.370, *p* = 0.051; Figure 4B). Moreover, the association between the anti-CMV IgG titer and the peripheral blood immune cell profile was investigated. We found that the anti-CMV IgG titer was positively correlated with the late-differentiated CD8+ T cell subset percentage (ρ = 0.589, *p* = 0.004) (Figure 4C).

## 4. Discussion

MS is a complex immune-mediate disorder characterized by the abnormal activation of immune cells which attack myelin in the CNS. Previous studies reported that CMV may have a detrimental or a beneficial role in MS, in both clinical and experimental settings, and can contribute to MS directly, through different mechanisms such as molecular mimicry and epitope spreading, or indirectly, via the activation and the expansion of specific immune cells [7,32]. Despite a few decades of investigation, the association between CMV and MS remains inconclusive. Together with other studies, we speculated on the detrimental role of CMV and its involvement in MS etiology and progression [33,34]. 

In line with other studies, we found no differences in the phenotype of total NK cells, CD56^bright^ cells, and CD56^dim^ cells in untreated MS patients compared to HDs [35]. Concerning T cells, MS patients did not differ from HDs in terms of percentages of CD8+ T cells and their subsets, including early-differentiated CD8+ T cells and late-differentiated CD8+ T cells. Otherwise, in line with previous reports, the MS patients presented lower levels of total T cells compared to the HDs [36]. 

Here, we found a significant increase in NKG2C expression levels on CD56^dim^CD57+ cells in the MS patients compared to the HDs, independently of their CMV serostatus. Importantly, higher NKG2C levels were positively associated with EDSS, suggesting that MS patients expressing higher levels of the NKG2C receptor are characterized by disease progression and the accumulation of disability over time. We observed a significant increase in NKG2C levels in CMV-seropositive MS patients compared to CMV-seronegative MS patients. In contrast, we did not find significant differences between CMV-seropositive and CMV-seronegative HDs, suggesting that the CMV-related increase in NKG2C was more pronounced in the MS patients. Moreover, our results showed that the NKG2C levels were increased in CMV-seropositive MS patients compared to CMV-seropositive HDs. These findings may identify some differences that could prove useful in MS monitoring. Nevertheless, they need to be confirmed in a larger group of patients to increase the power of the study.

Until the last decade, NK cells have largely been ignored in the MS field, differently from T and B cells, which have been recognized for their involvement in MS pathogenesis [37]. However, the involvement of NK cells remains controversial. Previous studies supported both deleterious and beneficial roles of NK cells in experimental autoimmune encephalomyelitis (EAE) [38]. The activation of NK cells is regulated by a balance between inhibitory and activating signals derived from membrane receptors, which interact with several ligands on their target cells [39]. Once activated, NK cells can induce apoptosis or release cytotoxic granules containing perforin and granzymes. The involvement of NK cells in MS pathogenesis is attributed to their ability to induce glial damage [40]. NKG2C is an activating receptor that plays an important role in promoting NK activation. This receptor interacts with non-classical HLA-E molecules of major histocompatibility (MHC) class I proteins expressed on the cell surface for triggering the NK cell cytotoxicity [41,42,43]. HLA-E molecules may be highly expressed in oligodendrocytes and microglia, as observed in active demyelinating lesions and, so, can be targeted by NK cells through NKG2C interaction [44,45]. Zaguia et al. performed immunohistochemical analyses on MS brain samples and found that HLA-E and NKG2C receptor are expressed in MS brain tissues [24]. 

The mechanisms by which CMV drives the expression of NKG2C on NK subsets are unknown; however, it is well established that NKG2C+ cells play a role in controlling CMV infection, both in vitro and in vivo [16]. Our findings suggest that CMV could act as a trigger in promoting an increase in the NKG2C expression levels, which may be deleterious in MS patients. We speculate that the NKG2C receptor might contribute to a self-directed attack on myelin resulting in CNS injury.

NK cells are critical for the control of viral infection and reactivation, and it is well known that CMV has a broad impact on their phenotype and function [46]. During CMV infection, the levels of the most differentiated and cytotoxic NK cells, CD56^dim^CD57+ (NK CD57+) cells, which exert a crucial role in CMV-infected cells, are increased [47]. These cells express high levels of CD57 receptor, a useful marker of NK maturation. In our study, the percentages of the most differentiated cell subset, i.e., NK CD57+ cells, were increased in CMV-seropositive MS patients compared to CMV-seronegative patients. We found no differences in terms of percentages for the most immature CD56^bright^ cells and the intermediate CD56^dim^ cells in CMV-seropositive MS patients compared to CMV-seronegative MS patients [48]. 

CD8+ T cells, together with NK cells, are essential for the control of viral infections. CMV exerts a persistent stimulation on CD8+ T cells, resulting in the development of a highly differentiated subset that is incapable of cell division. The advanced differentiation state is characterized by the acquisition of the CD57 marker and the lack of expression of the costimulatory receptor CD28, which is inversely expressed in early-differentiated CD8+ T cells [49]. The impact of late-differentiated CD8+ T cells on MS is not yet clear. Mikulkova et al. reported quantitative changes in the late-differentiated CD8+ T cell population in MS patients, considering these cells as immunosuppressive, although the expression of immunosuppressive markers was not evaluated in their study [28]. Conversely, in our study, the late-differentiated CD8+ T cell percentages did not differ between the MS patients and the HDs, suggesting that this population is not implicated in MS. Moreover, stratifying the population based on the serostatus, we found that CMV infection was associated with a significant increase in late-differentiated CD8+ T cells both in the MS patients and in the HDs [50]. These data confirmed that CMV may induce the accumulation of late-differentiated CD8+ T cells, both in health and in disease states. Moreover, these findings showed that CMV is a major viral driver of immunosenescence, and the expansion of this subset is a result of a constant stimulation by persistent antigens. Here, we also found that in MS patients, higher percentages of late-differentiated CD8+ T cells were associated with higher levels of anti-CMV antibodies. Immunosenescence may be associated with increased levels of anti-CMV antibodies and may be indicative of a poor control of the infection due to the worsening immunological status [51]. In contrast, we did not observe the same correlation in HDs. We hypothesize that MS patients may suffer from a high number of CMV reactivations and consequently exhibit increased interactions between the virus and CD8+ T cells, which become anergic and late-differentiated following the repeated antigenic stimulation. In our study, the MS patients with higher percentages of late-differentiated CD8+ T cells presented higher percentages of NK CD57+ cells, suggesting that CD57 expression is coordinately regulated as the immune system matures due to the persistent antigenic stimulation induced by CMV infection.

In addition to NK cells and CD8+ T cells, CD3+CD56+ natural killer T (NKT)-like cells represent a heterogeneous subset that shares some morphological and functional characteristics with both NK and T cells and are particularly interesting due to their dual role in the immune system [52]. NKT-like cells represent the first line of defense against several viral pathogens such as the influenza A and chikungunya viruses [53,54,55]. Van Dommelen et al. [56] demonstrated in the murine CMV model (MCMV) that NKT-like cells induce an antiviral state through the secretion of important cytokines, enhancing the immune response against MCMV. In line with McKay et al. [57], we did not find any differences in the percentages of NKT-like cells between MS patients and HDs. However, we observed a significantly higher percentage of NKT-like cells in CMV-seropositive MS patients compared to those who were CMV-seronegative. To the best of our knowledge, this is the first study investigating NKT-like cell expansion in CMV-seropositive MS patients. In conclusion, as we previously observed for other cell subsets, the increased percentage of NKT-like cells in MS patients may be explained by their protective role against CMV in preventing viral reactivation.

## 5. Conclusions

Cytomegalovirus, a ubiquitous herpesvirus, has long been studied for its putative association with MS, in both disease progression and pathogenesis. Due to its prolonged interaction with the immune system, it seems reasonable that this virus may impact MS at various levels. However, there are controversial reports on the association between CMV infection and MS. Our findings showed that CMV infection shapes the receptor repertoire of the immune system, both in healthy subjects and in MS patients. However, we identified some differences that could prove useful in MS monitoring. In this work, we hypothesized that CMV may contribute to immune-mediated processes, increasing the risk of disability worsening in patients showing a CMV-driven NKG2C receptor increase. This receptor could represent a pathogenic contribution, exacerbating the inflammatory response and consequently influencing the worsening of MS. However, we can only speculate a causal relationship between CMV infection and the risk of disability progression. Our results provide a starting point to identify specific biomarkers of disease progression, because our research was based on a limited number of samples, and our findings must be validated in larger studies. Further work is needed to understand the involvement of CMV in the multifaceted pathological picture of MS.

## Figures and Tables

**Figure 1 biology-13-00154-f001:**
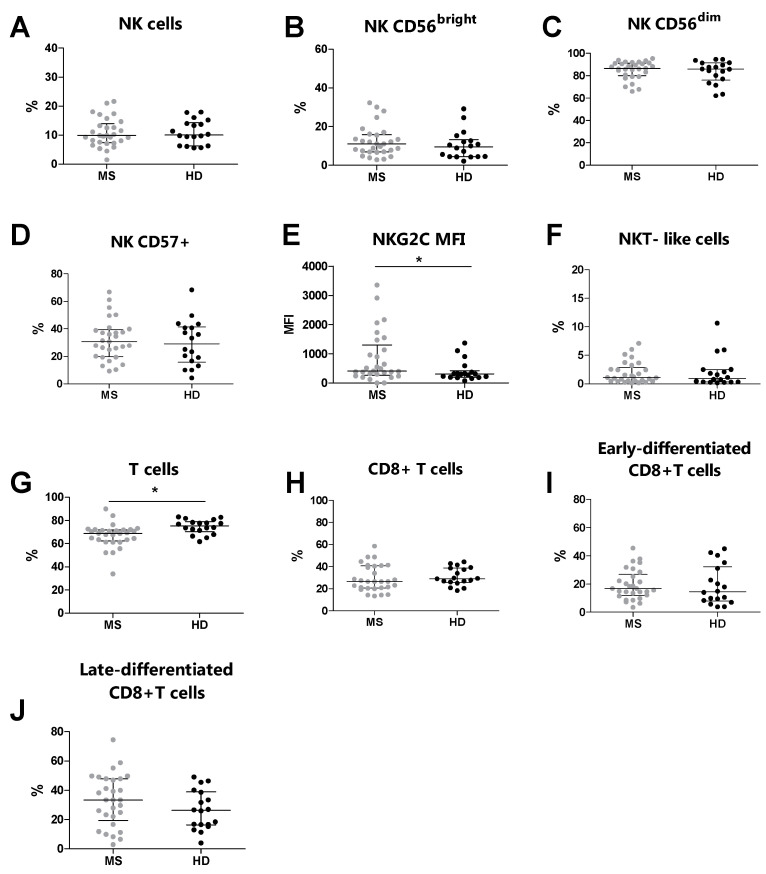
Immunophenotyping analysis was performed in MS patients (grey dots) and HDs (black dots). (**A**) The total % of NK cells, (**B**) % of CD56^bright^ NK cells, (**C**) % of CD56^dim^ NK cells, (**D**) % of CD57+ NK cells, (**E**) MFI of NKG2C on CD57+, (**F**) % of NKT-like cells, (**G**) total % of T cells, (**H**) % of CD8+ T cells, (**I**) % of early-differentiated CD8+ T cells, (**J**) % of late-differentiated CD8+ T cells. *p* < 0.05 was considered significant (*: *p* < 0.05). The data are shown as the median and IQR. MS: multiple sclerosis; HDs: healthy donors; MFI: median fluorescence intensity.

**Figure 2 biology-13-00154-f002:**
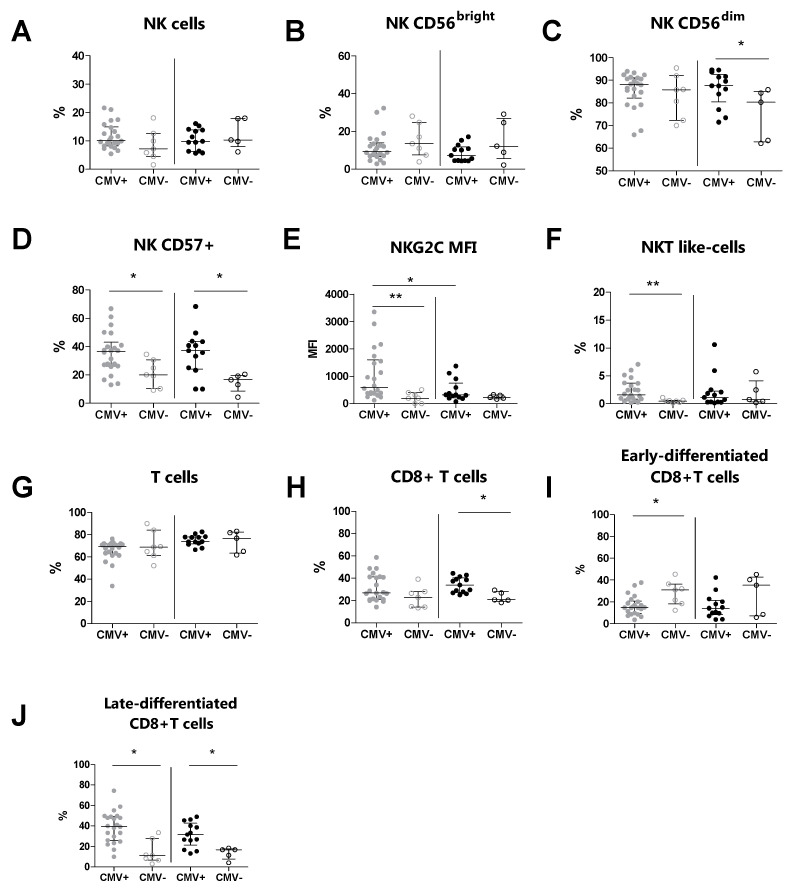
Immunophenotyping analysis was performed in MS patients (grey dots) and HD subjects (black dots) stratified for their CMV serostatus. (**A**) The total % of NK cells, (**B**) % of CD56^bright^ NK cells, (**C**) % of CD56^dim^ NK cells, (**D**) % of CD57+ NK cells, (**E**) MFI of NKG2C on CD57+, (**F**) % of NKT-like cells, (**G**) total % of T cells, (**H**) % of CD8+ T cells, (**I**) % of early-differentiated CD8+ T cells, (**J**) % of late-differentiated CD8+ T cells. *p* < 0.05 was considered significant; *: *p* < 0.05, **: 0.01 < *p* < 0.001. The data are shown as median and IQR. CMV+: CMV-seropositive; CMV−: CMV-seronegative; MS: multiple sclerosis; HDs: healthy donors; MFI: median fluorescence intensity.

**Figure 3 biology-13-00154-f003:**
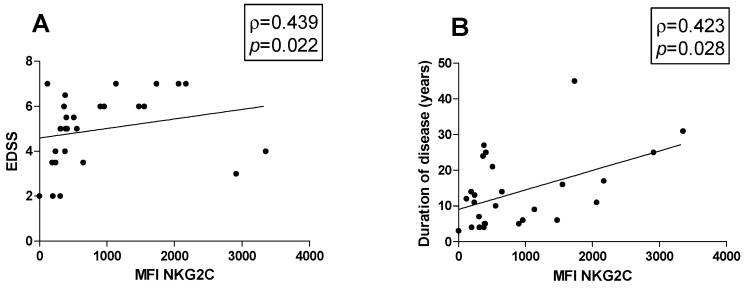
Correlations. Statistically significant correlations (*p* < 0.05) in MS patients between (**A**) EDSS and MFI value of NKG2C and between disease duration in years and (**B**) MFI of NKG2C. Data are shown as regression lines with the number of patients. The Spearman’s correlation coefficient ρ (Rho) and the *p* values are shown in the graphs.

**Figure 4 biology-13-00154-f004:**
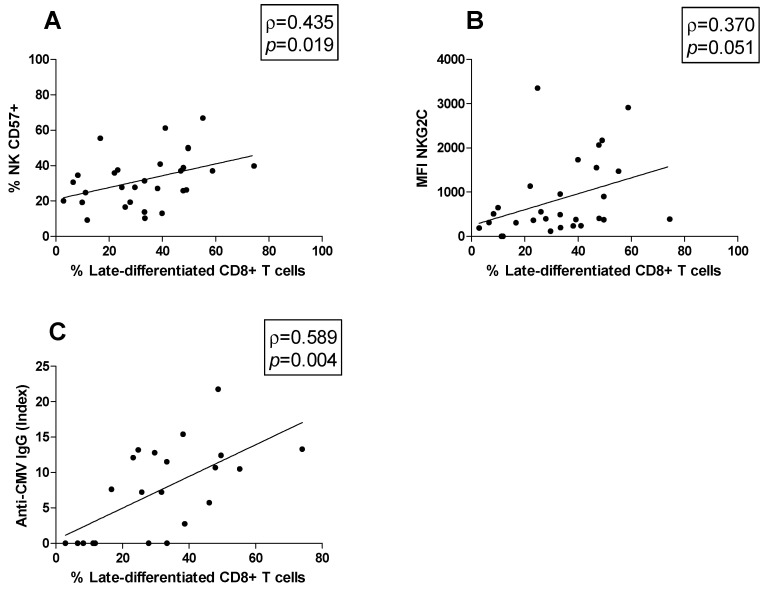
Correlations. Statistically significant correlations (*p* < 0.05) in MS patients between (**A**) the percentage of late-differentiated CD8+ T cells and that of CD57+ NK cells, (**B**) the percentage of late-differentiated CD8+ T cells and the MFI of NKG2C, (**C**) the percentage of late-differentiated CD8+ T cells and the anti-CMV IgG titer. Data are shown as regression lines with the number of patients. The Spearman’s correlation coefficient ρ (Rho) and the *p* values are shown in the graphs.

**Table 1 biology-13-00154-t001:** Demographics and clinical characteristics of the study population.

	MS	HDs
Characteristics		
Total	74	18
Gender, *n* (%)		
Male, *n* (%)	37 (50)	7 (39)
Female, *n* (%)	37 (50)	11 (61)
Age, median (IQR)	51 (43–58)	52 (38–59)
Disease duration, median (IQR)	11 (6–19)	-
EDSS, median (IQR)	5.0 (3.0–6.0)	-
CMV serostatus, *n* (%)		
IgG+/IgM−, *n* (%)	51 (69)	13 (73)
IgG−/IgM−, *n* (%)	23 (31)	5 (27)
IgG+/IgM+, *n* (%)	0	0
CMV DNA, *n* (%)		
Positive, *n* (%)	12 (16)	0
Negative, *n* (%)	62 (84)	0

MS: multiple sclerosis; HDs: healthy donors; IQR: interquartile range; EDSS: Expanded Disability Status Scale.

**Table 2 biology-13-00154-t002:** Distribution of anti-CMV IgG and presence of CMV DNA among MS patients according to their demographic and clinical characteristics.

	Anti CMV-IgG		*p* ^a^	CMV DNA		*p* ^b^
	Positive	Negative		Positive	Negative	
Total MS, *n* (%)	51 (69)	23 (31)	-	12 (16)	62 (84)	-
Male, *n* (%)	26 (70)	11 (30)	-	6 (16)	31 (84)	-
Female, *n* (%)	25 (67)	12 (33)	-	6 (16)	31 (84)	-
Age, median, (IQR)	52 (46–59)	44 (35–54)	**0.02**	53 (41–56)	50 (43–58)	0.91
Disease duration, median (IQR)	11 (6–18)	8 (3–19)	0.32	11 (8–24)	11 (5–18)	0.28
EDSS, median (IQR)	5.0 (3.5–6.0)	5.0 (1.0–6.0)	0.25	5.5 (2.0–6.5)	5.0 (3.0–6.0)	0.77

MS: multiple sclerosis; IQR: interquartile range; EDSS: Expanded Disability Status Scale. ^a^ The Mann–Whitney U test was used to compare medians between CMV-seropositive (anti-CMV IgG positive) and CMV-seronegative (anti-CMV IgG negative) MS patients. ^b^ The Mann–Whitney U test was used to compare medians between CMV DNA-positive MS patients and CMV DNA-negative MS patients. Differences were considered statistically significant when *p* < 0.05 (in bold).

**Table 3 biology-13-00154-t003:** Comparison between peripheral blood immune cell profiles in MS patients and HDs.

	MS (*n* = 29)	HDs (*n* = 18)	*p* ^a^
NK cells	9.9 (7.4–13.9)	10.1 (6.3–14.3)	0.818
CD56^bright^ cells	11.0 (7.0–15.9)	9.5 (4.4–13.4)	0.352
CD56^dim^ cells	86.5 (79.9–91.2)	85.9 (76.1–91.5)	0.614
CD57+ cells	30.6 (19.7–39.3)	29.1 (15.7–41.4)	0.662
NKG2C (MFI)	409 (274–1304)	318 (195.5–424)	**0.041**
NKT-like cells	1.1 (0.5–2.9)	1.0 (0.3–2.5)	0.661
T cells	68.9 (62.3–71.3)	75.3 (70.4–78.9)	**0.001**
CD8+ cells	26.5 (20.7–40.7)	29.2 (25.8–38.7)	0.376
Early-differentiated CD8+ cells	16.8 (11.8–26.9)	14.5 (8.01–32.1)	0.592
Late-differentiated CD8+ cells	33.3 (19.4–47.9)	26.3 (16.3–39.0)	0.208

MS: multiple sclerosis; HDS: healthy donors; MFI: median fluorescence intensity. ^a^ The Mann–Whitney U test was used to compare medians between MS patients and HDs. Data are expressed as the median (interquartile range, IQR) of cell percentages. Differences were considered statistically significant when *p* < 0.05 (in bold).

**Table 4 biology-13-00154-t004:** The peripheral blood immune cell profiles. MS patients and HDs were stratified according to their CMV serostatus and classified as either CMV-seropositive or CMV-seronegative.

	MS (*n* = 29)			HDs (*n* = 18)		
	CMV+ (*n* = 22)	CMV− (*n* = 7)	*p* ^a^	CMV+ (*n* = 13)	CMV− (*n* = 5)	*p* ^b^
NK cells	10.1 (8.2–14.9)	7.2 (4.5–14.3)	0.153	9.8 (6.3–13.9)	10.3 (7.9–17.9)	0.430
CD56^bright^ cells	9.3 (6.8–13.9)	13.5 (7.5–24.7)	0.320	7.2 (4.4–11.8)	12.10 (5.5–26.9)	0.324
CD56^dim^ cells	88.1 (82.1–91.1)	85.8 (72.3–92)	0.558	87.6 (80.5–92.6)	80.3 (62.7–85.1)	**0.038**
CD57+ cells	36.5 (26.2–43.1)	20.0 (10.3–30.1)	**0.020**	37.1 (24.2–43.6)	16.6 (8.6–19.8)	**0.018**
NKG2C (MFI)	600.5 (375.5–1597)	197 (0–399)	**0.008**	320 (210–750)	224 (176–321)	0.278
NKT-like cells	1.6 (0.8–3.7)	0.5 (0.2–0.6)	**0.008**	1.1 (0.3–2.3)	0.8 (0.3–4.1)	1.000
T cells (%)	69.5 (62.8–71.7)	68.9 (61.2–84.1)	0.939	73.8 (71.9–78.1)	76.8 (63.4–82.4)	1.000
CD3+CD8+ cells	27.0 (21.1–41.3)	22.8 (14.2–28.1)	0.101	34.0 (27.4–40.3)	20.7 (19.3–28.1)	**0.014**
Early-differentiated CD8+ cells	14.7 (9.43–20.6)	30.1 (18.2–36.3)	**0.023**	14.1 (8.2–21.4)	35.2 (7.1–42.7)	0.375
Late-differentiated CD8+ cells	39.5 (25.8–49.2)	11.2 (6.5–27.9)	**0.002**	31.6 (21.3–42.7)	16.7 (7.6–17.6)	**0.023**

CMV+: CMV-seropositive; CMV−: CMV-seronegative; MFI: median fluorescence intensity. ^a^ The Mann–Whitney U test was used for comparing CMV+ MS patients and CMV− MS patients. ^b^ The Mann–Whitney U test was used for comparing CMV+ HDs and CMV− HDs. Data are expressed as the median (interquartile range, IQR) of cell percentages. Differences were considered statistically significant when *p* < 0.05 (in bold).

## Data Availability

The original contributions presented in the study are included in the article. Further inquiries can be directed to the corresponding author.

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
