# Peer review of "The Impact of Cytomegalovirus Infection on Natural Killer and CD8+ T Cell Phenotype in Multiple Sclerosis"

_biology, 2024, doi:10.3390/biology13030154_

Round 1

Reviewer 1 Report

Comments and Suggestions for Authors

Overall, the manuscript reports and association of NKG2C+ ‘adaptive’ NK cells, which are expanded in CMV infection, with a poor ‘Expanded Disability Status Scale’ (EDSS). I am not an expert in MS and the authors highlight controversy in the MS field regarding the association of ‘adaptive NK’ with the EDSS e.g. some manuscripts report an association with improved EDSS and whilst others report associations with a poor EDSS. I found the current paper does little to resolve this controversy perhaps reflected by the superficial analysis of NK subsets.

Minor points:

Fig 1. Unless there is primary data to report from this figure, move ‘Gating strategy’ to supplementary data

Fig. 3 appears to be an extension of Fig. 2.

‘EDSS’ should be defined in the Abstract.

Comments on the Quality of English Language

English is poor and needs some editing.

Author Response

Comments and Suggestions for Authors

Overall, the manuscript reports and association of NKG2C+ ‘adaptive’ NK cells, which are expanded in CMV infection, with a poor ‘Expanded Disability Status Scale’ (EDSS). I am not an expert in MS and the authors highlight controversy in the MS field regarding the association of ‘adaptive NK’ with the EDSS e.g. some manuscripts report an association with improved EDSS and whilst others report associations with a poor EDSS. I found the current paper does little to resolve this controversy perhaps reflected by the superficial analysis of NK subsets.

Author response:

We are grateful for your comments on the manuscript. According to your advice, we have revised the manuscript. Our responses to your comments appear below. Moreover, we have carefully revised our manuscript for improve the English language.

Minor points:

1-Fig 1. Unless there is primary data to report from this figure, move ‘Gating strategy’ to supplementary data

1- Author response:

Thank you for your valuable advice. We have added the suggested content to the manuscript on Supplementary Materials.

2-Fig. 3 appears to be an extension of Fig. 2.

2-Author response:

We are grateful for your feedback. Flow cytometry immunophenotypic analysis was performed in MS and HDs, and the results are summarized in Figure 2. In the second step, MS patients and HDs were stratified according to their CMV serostatus and classified as either CMV-seropositive or CMV-seronegative, as showed in Figure 3. We think that Figure 3 is useful for observing CMV-associated immune cell alterations in MS and HDs.

3-EDSS should be defined in the Abstract

3- Author response:

Thank you for your valuable advice. We have added the following sentences to the Abstract: “…Expanded Disability Status Scale (EDSS), a clinical scale that measures disease progression”.

Reviewer 2 Report

Comments and Suggestions for Authors

In this manuscript, Perri et al. studied the role played by cytomegalovirus (CMV)-induced immune responses on the course of multiple sclerosis (MS). By using flow cytometry analysis, the authors have found that the expression of the activating immunoreceptor NKG2C on CD57+ NK cells and the presence of late/senescent CD8+ T lymphocytes are both characteristic of a subpopulation of MS human patients infected with CMV, suggesting a detrimental impact of CMV infection in MS.

Major comment:

The work by Perri et al. is interesting and well performed, although of limited novelty taking into account the previous accumulative evidence of the prevalence of CMV in patients with MS and the contribution of CMV-directed T cell subsets to MS disease. Mechanistic insights helping to understand how CMV infection promotes MS are required to improve the interest and novelty of the work. In this regard, despite the significant correlation among lymphocyte subsets, NKG2C expression and MS, whether NKG2C expression on NK cells may affect MS disease progression remains an important open question that should be addressed before the manuscript may be considered for publication.

Some specific points:

1.- Discussion, line 327: The authors mention that in some cases CMV infection can cause a beneficial role in MS disease. May the authors deal with this subject in greater depth on the Introduction section?

2.- Figure 3: Compared with NKG2C expression on CMV+ healthy donors, increased expression of NKG2C on cd57+ NK cells from CMV+ MS patients is significant, clear and interesting (panel E). However, the differences showed for late-differentiated CD8+ T lymphocytes are much less convincing as they seem associated to both CMV-infected MS and HD patients (panel L). Do late-differentiated T lymphocytes actually play a role in MS disease? This issue should be convincingly addressed.

3.- Is NKG2C expression on CD57+ NK cells involved in MS disease progression or is it merely a consequence? May NKG2C expression be used as a predictive biomarker of the progression of MS disease in CMV infected patients? Moreover, If NKG2C expression levels can contribute to tissue injury, such as it has been speculated by the authors (line354-355), why NKG2C expression correlates fairly well with the duration of disease? May it represent a beneficial parameter?

4.- Figure 4: Opposite correlation of NK CD56dim and NKCD56bright cells with duration of MS disease is shown (panels C and D, respectively). Would these NK cells subsets be useful as predictive cell biomarkers of MS progression?

Minor point:

5.- The manuscript must be extensively revised, and typographic errors corrected.

Author Response

Major comment:

The work by Perri et al. is interesting and well performed, although of limited novelty taking into account the previous accumulative evidence of the prevalence of CMV in patients with MS and the contribution of CMV-directed T cell subsets to MS disease. Mechanistic insights helping to understand how CMV infection promotes MS are required to improve the interest and novelty of the work. In this regard, despite the significant correlation among lymphocyte subsets, NKG2C expression and MS, whether NKG2C expression on NK cells may affect MS disease progression remains an important open question that should be addressed before the manuscript may be considered for publication.

Author response:

Thanks for the comments that help to improve the quality of the document.. Based on your comments and suggestions, we have made extensive revisions to the manuscript. We add more introduction (line 60-72) and discussion concerning the role of NKG2C receptor on NK cells (line 326-341). We also highlighted the limit of the study, poorly considered in the previous version of manuscript (line 319-321; line 404-407).

Specific points:

1-Discussion, line 327: The authors mention that in some cases CMV infection can cause a beneficial role in MS disease. May the authors deal with this subject in greater depth on the Introduction section?

1-Author response:  

We are sorry for missing informations. The introduction has been modified as suggested (line 48-51).

2-Figure 3: Compared with NKG2C expression on CMV+ healthy donors, increased expression of NKG2C on cd57+ NK cells from CMV+ MS patients is significant, clear and interesting (panel E). However, the differences showed for late-differentiated CD8+ T lymphocytes are much less convincing as they seem associated to both CMV-infected MS and HD patients (panel L). Do late-differentiated T lymphocytes actually play a role in MS disease? This issue should be convincingly addressed.

2- Author response:

Thank you for your valuable suggestion. The discussion has been modified on line 356-370. In MS the role of late-differentiated CD8+ T cell is not well delucidated. A study of Mikulkova et al., reported quantitative changes of late-differentiated CD8+ T cell population in MS and considered these cells as immunosuppressive, although the expression of immunosuppressive markers was not evaluated in their research [1]. In our study, we did not find significant differences of late-differentiated CD8+ T cell percentage in MS and HDs, suggesting that this population was not implicated in MS. On the other hand, we found that CMV infection induced an increase of late-differentiated CD8+ T-cells both in MS patients and HDs. These data confirmed that CMV may induce the accumulation of late-differentiated CD8+ T-cells, both in health than in disease. The progressive accumulation of this population is a result of lifetime exposure to persistent antigens, and we support that CMV infection is the major viral driver of the expansion of this subset.

3-Is NKG2C expression on CD57+ NK cells involved in MS disease progression or is it merely a consequence? May NKG2C expression be used as a predictive biomarker of the progression of MS disease in CMV infected patients? Moreover, If NKG2C expression levels can contribute to tissue injury, such as it has been speculated by the authors (line354-355), why NKG2C expression correlates fairly well with the duration of disease? May it represent a beneficial parameter?

3- Author response:

Thank you for your valuable suggestion. The text has been modifing on line 323-337 of discussion. In our study we found a significantly increase of NKG2C expression levels on CD56dimCD57+ cells in MS patients compared to HDs, and observed a positive correlation between that NKG2C levels and EDSS score, a clinical parameter which describe severity of disability in patients with MS. These findings suggest a pathogenic contribution to MS, causing inflammatory lesions and exacerbating the inflammatory response [2]. Moreover, we observed a significant increase of NKG2C levels in CMV-seropositive MS patients compared to CMV-seronegative MS patients. In contrast, no significant differences were found in HDs, between CMV-seropositive and CMV-seronegative counterparts, suggesting that the CMV-related expression of NKG2C was more pronounced in MS. Moreover, our results showed that NKG2C levels were increased between CMV-seropositive MS patients compared to CMV-seropositive HDs, underlining the important involvement of CMV in MS disease. Our findings suggest that CMV could act as a trigger in promoting the increase of NKG2C expression levels, which may be deleterious in MS patients. However, our results need to be confirmed on a larger group of patients to increase the power of the study and to approve NKG2C as an indicator of MS progression.

In our study the duration of the disease is not indicative of life expectancy, rather would be indicative of disease progression. We did not find correlations between EDSS severity and duration of disease, however, these association can be observed in study with a large sample size [3,4]. A large study of the Swiss Multiple Sclerosis Registry [5] demonstrated that MS duration was associated with having severe disability. This might be expected, since people with MS with a longer disease course show an accumulation of disability. To validate this finding would be useful to performe a longitudinal evaluation of NKG2C levels in MS patients.

4-Figure 4: Opposite correlation of NK CD56dim and NKCD56bright cells with duration of MS disease is shown (panels C and D, respectively). Would these NK cells subsets be useful as predictive cell biomarkers of MS progression?

4- Author response:

Thanks for your comments. In line with other studies, we found no differences in the phenotype of CD56bright cells and CD56dim cells in untreated MS patients compared to HDs. Furthermore, we showed that CMV infection was not associated with alteration on CD56bright cell and CD56dim cell phenotype (except for CD56dim cells in HDs). Overall, our findings did not suggest a possible involvement of CD56bright cells and CD56dim cells in MS disease, but they were worth mentioning. In contrast, we found that CD56bright cells and CD56dim cells were associated to disease duration, with a negative correlation and a positive correlation, respectively. After a evaluation we believe that these correlations do not make a significant contribution to the study, therefore we believe it is more appropriate not to report them. We are sorry if these changes were applied after the manuscript was submitted.

References

  1. Mikulkova, Z.; Praksova, P.; Stourac, P.; Bednarik, J.; Strajtova, L.; Pacasova, R.; Belobradkova, J.; Dite, P.; Michalek, J. Numerical Defects in CD8+CD28- T-Suppressor Lymphocyte Population in Patients with Type 1 Diabetes Mellitus and Multiple Sclerosis. Cell Immunol 2010, 262, 75–79, doi:10.1016/j.cellimm.2010.02.002.
  2. Zaguia, F.; Saikali, P.; Ludwin, S.; Newcombe, J.; Beauseigle, D.; McCrea, E.; Duquette, P.; Prat, A.; Antel, J.P.; Arbour, N. Cytotoxic NKG2C+ CD4 T Cells Target Oligodendrocytes in Multiple Sclerosis. J Immunol 2013, 190, 2510–2518, doi:10.4049/jimmunol.1202725.
  3. Trojano, M.; Liguori, M.; Bosco Zimatore, G.; Bugarini, R.; Avolio, C.; Paolicelli, D.; Giuliani, F.; De Robertis, F.; Marrosu, M.G.; Livrea, P. Age-Related Disability in Multiple Sclerosis. Annals of Neurology 2002, 51, 475–480, doi:10.1002/ana.10147.
  4. Manouchehrinia, A.; Westerlind, H.; Kingwell, E.; Zhu, F.; Carruthers, R.; Ramanujam, R.; Ban, M.; Glaser, A.; Sawcer, S.; Tremlett, H.; et al. Age Related Multiple Sclerosis Severity Score: Disability Ranked by Age. Mult Scler 2017, 23, 1938–1946, doi:10.1177/1352458517690618.
  5. Stanikić, M.; Salmen, A.; Chan, A.; Kuhle, J.; Kaufmann, M.; Ammann, S.; Schafroth, S.; Rodgers, S.; Haag, C.; Pot, C.; et al. Association of Age and Disease Duration with Comorbidities and Disability: A Study of the Swiss Multiple Sclerosis Registry. Multiple Sclerosis and Related Disorders 2022, 67, 104084, doi:10.1016/j.msard.2022.104084.

Reviewer 3 Report

Comments and Suggestions for Authors

This paper by Perri et al. analyzes the effect of chronic Cytomegalovirus (CMV) infection on the pathogenesis of Multiple Sclerosis(MS), focusing on NK cells and T cells. The results of this study do not reveal a causal relationship between CMV infection and MS disease progression, the authors found that NKG2C expression was higher in CD57+ NK cells of MS patients who were CMV seropositives than CMV seronegatives, and that the proportion of CD3+CD56+ NKT-like cells was increased in CMV seropositve MS patients. These findings are novel and may have clinical implications for MS pathogenesis.

Major point:

1. In Fig. 3E, the authors show that NKG2C expression is significantly enhanced in CMV seropositive MS patients than in CMV seronegative MS patients, but is this enhancement absent in HD? Can the authors explain why NKG2C is not upregulated in HD, even though CD57 expression is enhanced? Could this be inconsistent with previous reports (PNAS, 108: 14725-14732, 2011) that CMV infection increases the CD57+NKG2C cell population? Is it possible that there is an insufficient number of CMV seropositive specimens in HD? Is there any possibility that the number of CMV seropositive samples in HD is insufficient? The same may apply to the case of NKT like-cells in 3F. The authors should discuss these possiblities.

2. It would be helpful to the reader if there is an explanation of how to select samples for CMV+ MS patients (n=22) in Table 4.

Author Response

Comments to revisions:

This paper by Perri et al. analyzes the effect of chronic Cytomegalovirus (CMV) infection on the pathogenesis of Multiple Sclerosis (MS), focusing on NK cells and T cells. The results of this study do not reveal a causal relationship between CMV infection and MS disease progression, the authors found that NKG2C expression was higher in CD57+ NK cells of MS patients who were CMV seropositives than CMV seronegatives, and that the proportion of CD3+CD56+ NKT-like cells was increased in CMV seropositve MS patients. These findings are novel and may have clinical implications for MS pathogenesis

Author response:

The authors thank the reviewer for the positive comments. In this research, we explored the impact of latent CMV infection on MS through a characterization of peripheral NK cells and T cells in MS, to provide a better understanding of immune cell alterations CMV-associated and to investigate its putative involvement on MS disease. Our results are in line with several evidences which indicate that CMV infection can be associated with alteration of NK and T cells phenotype. Importantly, we highlighted an increase expression of CMV-induced NKG2C levels in MS patients compared to HDs, identifying a difference that may prove useful in monitoring MS. However, we can only speculate a causal relationship between CMV infection and the worsening of MS progression. In fact, our results provide a starting point to identify specific biomarkers of progression, because this research is based on a limited number of samples, and findings must be validated in larger studies.

Major points:

1-In Fig. 3E, the authors show that NKG2C expression is significantly enhanced in CMV seropositive MS patients than in CMV seronegative MS patients, but is this enhancement absent in HD? Can the authors explain why NKG2C is not upregulated in HD, even though CD57 expression is enhanced? Could this be inconsistent with previous reports (PNAS, 108: 14725-14732, 2011) that CMV infection increases the CD57+NKG2C cell population? Is it possible that there is an insufficient number of CMV seropositive specimens in HD? Is there any possibility that the number of CMV seropositive samples in HD is insufficient? The same may apply to the case of NKT like-cells in 3F. The authors should discuss these possiblities.

1- Author response:

Thanks for your comments. In our study, we found a significant increase of NKG2C levels in CMV seropositive MS patients compared to CMV seronegative MS patients. In contrast, we did not find significant differencies between CMV seropositive and CMV-seronegative counterparts in HDs, suggesting that the CMV-related increase of NKG2C was more pronounced in MS. These findings highlight an uncoupled expression of CMV-induced NKG2C increase in MS patients compared to HDs, identifying a difference that may prove useful in monitoring MS. However, this results might be in contrast with Gumà and collegues which first reported the observation that NK cells expressing NKG2C were increased in CMV-seropositive HDs. In our study, althought no significantly, we observed a trend for increased levels of NKG2C in CMV-seropositive HDs compared to CMV seronegative HDs. However, it is conceivable that the increasing of HDs samples could enhance the statistically significance of results.

2-It would be helpful to the reader if there is an explanation of how to select samples for CMV+ MS patients (n=22) in Table 4.

2- Author response:

Thanks for your suggestion. The table and the text were modified as suggested (line 253).

Round 2

Reviewer 2 Report

Comments and Suggestions for Authors

The authors have satisfactorily addressed all my major concerns. However, they should revise their manuscript more thoroughly. The new version of the manuscript includes important typographic errors that must be corrected, particularly the inclusion of the new references on the Introduction and Discussion sections.

Author Response

Thank you for raising important questions that helped improve the quality of the manuscript.

We agree with all your comments and the necessary corrections have been made. We have corrected the typographical errors and added new references to the manuscript. Furthermore, as suggested, we have provided a separate conclusion section.